# Burden of chronic kidney disease in the general population and high-risk groups in South Asia: A systematic review and meta-analysis

Nipun Shrestha[1]*, Sanju Gautam[2], Shiva Raj Mishra[3], Salim S. Virani[4], Raja Ram Dhungana[5]

1 Community Health Development Nepal, Kathmandu, Nepal, 2 Tokha Municipality, Kathmandu, Nepal, 3 Nepal Development Society, Bharatpur, Nepal, 4 Section of Cardiovascular Research, Baylor College of Medicine and Section of Cardiology, Michael E. DeBakey Veterans Affairs Medical Center, Houston, TX, United States of America, 5 Manmohan Memorial Institute of Health Sciences, Kathmandu, Nepal

* drnipunsth@gmail.com

**Data Availability Statement:** All relevant data are within the manuscript.

**Funding:** The authors received no specific funding for this work.

## Abstract

### Background

Chronic kidney disease (CKD) is an emerging public health issue globally. The prevalence estimates on CKD in South Asia are however limited. This study aimed to examine the prevalence of CKD among the general and high-risk population in South Asia.

### Methods

We conducted a systematic review and meta-analysis of population-level prevalence studies in South Asia (Afghanistan, Bangladesh, Bhutan, Maldives, Nepal, India, Pakistan, and Sri Lanka). Three databases namely PubMed, Scopus and Web of Science were systematically searched for published reports of kidney disease in South Asia up to 28 October 2020. A random-effect model for computing the pooled prevalence was used.

### Results

Of the 8749 identified studies, a total of 24 studies were included in the review. The pooled prevalence of CKD among the general population was 14% (95% CI 11–18%), and 15% (95% CI 11–20%) among adult males and 13% (95% CI 10–17%) in adult females. The prevalence of CKD was 27% (95% CI 20–35%) in adults with hypertension, 31% (95% CI 22–41%) in adults with diabetes and 14% (95% CI 10–19%) in adults who were overweight/obese. We found substantial heterogeneity across the included studies in the pooled estimates for CKD prevalence in both general and high-risk populations. The prevalence of CKD of unknown origin in the endemic population was 8% (95% CI 3–16%).

**Competing interests:** Nipun Shrestha serves as an academic editor at Plos One. Sanju Gautam, Shiva Raj Mishra and Raja Dhungana declare that no competing interests exists. Salim Virani: Research funding: US Department of Veterans Affairs, World Heart Federation, Tahir and Jooma Family Honorarium: American College of Cardiology (Associate Editor for Innovations, acc.org). This does not alter our adherence to PLOS ONE policies on sharing data and materials.

## Conclusion

Our study reaffirms the previous reports that CKD represents a serious public health challenge in South Asia, with the disease prevalent among 1 in 7 adults in South Asian countries.

## Introduction

Chronic kidney disease (CKD) is an emerging public health issue globally. The Global Burden of Disease study estimated about 1.4 million deaths globally from CKD in 2019, a 20% increase from 2010, one of the largest rises among the top causes of death [1]. CKD disproportionately impacts low and middle-income countries where both prevalence and deaths due to CKD are significantly higher [2, 3]. The increasing burden of CKD across the globe has been attributed to the meteoric rise in the prevalence of its risk factors such as obesity, hypertension, diabetes, and other cardiovascular diseases (CVD) [4, 5]. Multiple Risk Factor Intervention Trial (MRFIT) showed that CKD patients with hypertension, compared to non-hypertensive, have nearly 22 times higher risk of end-stage renal disease (ESRD) [6]. Further, co-morbid hypertension and/or diabetes among CKD patients may exacerbate the prognosis and results in higher mortality and cardiovascular events [4]. But in some regions of South Asian countries such as Sri Lanka and India, other causes such as environmental toxins (heavy metals, pesticides) and underground water with high fluoride levels have also been implicated in the high burden of CKD also known as a CKD of unknown origin (CKDu) [7].

Limited access to renal replacement therapy (dialysis or kidney transplantation) in South Asia lead to premature deaths among people with CKD who go onto develop ESRD [8]. Even for those who receive the renal replacement therapy, the financial burden that it inflicts is often catastrophic and poses far-reaching implications for individuals and their families [8, 9]. Strengthening screening services and managing underlying metabolic risk factors should be the priority in the low economy countries of South-Asia. Thus, updated evidence of the prevalence of CKD in South Asia are urgently warranted. In this systematic review, we assessed the burden of CKD in the general population and high-risk groups in South Asian countries that included Afghanistan, Bangladesh, Bhutan, Maldives, Nepal, India, Pakistan, and Sri Lanka.

## Methods

The review protocol has been published in PROSPERO: CRD42020220194 and has been conducted adhering to the Meta-analysis Of Observational Studies in Epidemiology (MOOSE) guidelines [10].

### Search strategy and selection criteria

We searched PubMed, Scopus and Web of Science for published reports of kidney disease in South Asia up to 28 October 2020. We used Boolean logic with search terms including "kidney disease", "renal function", "renal insufficiency", and "South Asia" (S1 Table). We also searched the reference list of included studies and systematic reviews on the similar topic identified in the database search.

**Inclusion and exclusion criteria.** The studies retrieved through database research were screened for eligibility independently by two authors (RD, SG), and any disagreements were resolved through discussion with the third author (NS). For inclusion, studies had to fulfill the following criteria:

1. Study population: Population or hospital-based, randomized controlled trial (RCT) or non-randomized studies including cross-sectional, cohort, and case-cohort studies that reported prevalence or provided data that allowed computation of the prevalence of CKD in adults (aged 18 and above), conducted in at least one of the South Asian Association for Regional Cooperation (SAARC) countries that included Afghanistan, Bangladesh, Bhutan, India, Maldives, Nepal, Pakistan in any age group or any gender.

2. Exposure: chronic kidney disease should have been defined as

    a. the presence of kidney damage (proteinuria) and/or

    b. stimated Glomerular filtration rate (eGFR) < 60 mL/min/1.73 m$^2$, by the Modification of Diet in Renal Disease (MDRD) and Chronic Kidney Disease Epidemiology Collaboration (CKD-EPI) formulae or a creatinine clearance of less than 60 mL per min by the Cockcroft-Gault formula.

These abnormalities should persist for three months for a diagnosis of CKD. However, for population-based studies, one measurement of kidney function is considered acceptable for the diagnosis of CKD. This definition is based on the NKF KDOQI (National Kidney Foundation Kidney disease outcomes quality initiative) or KDIGO (Kidney Disease: Improving Global Outcomes) guidelines [11, 12]. For CKDu, we relied on the definition reported by the included studies.

3. Comparison: For prevalence estimate of CKD/CKDu in the general population, those studies which reported the number of CKD cases and sample population were included. Similarly, for prevalence estimate of CKD in high-risk population, those studies which reported the number of CKD cases and high-risk population (hypertension, diabetes and overweight/obesity) were included.

4. Outcome: The included studies were required to report the prevalence of CKD/CKDu in the general population, CKD in high-risk population or provided data that allowed computation of the prevalence. In instances where there were multiple publications from the same dataset, we only included the most comprehensive article that reported the outcomes considered in this review.
Exclusion criteria: Studies were excluded if they had no criteria for the diagnosis of CKD, did not include prevalence or were in a specialist restricted population (e.g. acute hospital patient cohort, nursing home). For the prevalence of CKD/CKDu in the general population, we excluded studies having a sample size less than 500. Similarly, for summarizing the prevalence of CKD in high-risk population, studies with a sample size of less than 300 were excluded. The smaller sample size studies were excluded to avoid selection bias from small studies.

## Data extraction

Two authors (NS and SG) individually extracted data from the identified articles using a piloted data extraction table developed in excel for this review. This table included 1) author details: names and publication year, 2) Study characteristics: country, data collection period, setting, data source, sampling method, sample size, 3) Participant characteristics: mean age, gender, mean BMI, and 4) CKD characteristics: creatinine assessment method, eGFR equation, proteinuria assessment method, follow up, prevalence, number of participants tested and diagnosed with CKD overall and by subgroups of interest. The extracted data from both the authors were merged and later crosschecked by a third author (RD) to ensure consistency.

### Assessment of the risk of bias in included studies

Two reviewers (RD & SG) independently assessed the risk of bias in included studies, with disagreements being resolved by discussion or by consulting a third reviewer (NS). A checklist with a ten-item rating developed by Hoy et al. [13] was used for risk of bias assessment. The checklist assesses the methodological quality of the included studies in the following domains: sampling, the sampling technique and size, outcome measurement, response rate, and statistical reporting. The reviewers assigned a score of 1 (yes) or 0 (no), for each item. The overall quality score (range: 0–10) was generated by adding the scores for the individual item. The studies with an overall score of higher than 8 were judged at high quality, a score of 6–8 were judged as moderate quality and a score of 5 or lower were judged low quality.

### Statistical analysis

For each study, the unadjusted prevalence of CKD and standard errors were calculated (number of cases/sample size) based on the information on crude numerators and denominators provided in individual studies. The variances in the included studies was stabilised using the Freeman-Tukey Double Arcsine Transformation before estimating the pooled prevalence [14]. For the studies that reported eGFR from multiple equations, we used the Modification of Diet in Renal Disease (MDRD) then the Chronic Kidney Disease Epidemiology Collaboration (CKD-EPI) equation, and lastly the Cockcroft-Gault formula in the main analyses. We used the DerSimonian-Laird random-effects models to generate the pooled prevalence of CKD according to each diagnostic criterion. The random-effects model was chosen with the assumption that CKD prevalence estimates across the included studies would be variable. We also examined prevalence in high-risk populations (hypertension, diabetes mellitus and overweight/obese) and prevalence of CKDu using the random-effects model. The heterogeneity in the included studies was assessed using $I^2$ statistics, with $I^2$ values higher than 70% considered as evidence of substantial heterogeneity [15]. Subgroup analysis by country, gender, equation used for GFR estimation, and meta-regression by mean age, mean BMI and survey year was performed to identify the sources of heterogeneity across the studies. Publication bias was assessed by Egger's test and funnel plot for pooled analyses with 10 or more studies [16]. All calculations were conducted using STATA (StataCorp LLC, USA).

## Results

### Study selection

The search strategy yielded 8739 citations. An additional 10 studies were located from secondary searches. After removing duplicates, a total of 7439 studies were retrieved for the title and abstract screening, of which 47 studies were selected for full-text screening. Twenty-three studies were excluded for the following reasons: wrong study population, wrong outcomes, wrong study design, duplicate studies, the non-standard definition of CKD, sample size less than 500 and sample size less than 300 for studies conducted in high-risk groups. Finally, 24 studies [17–40] were included in this systematic review and meta-analysis (Fig 1). Two studies reported prevalence of low GFR and proteinuria but did not report CKD prevalence [33, 35]. Out of 22 studies, 15 studies reported CKD prevalence in the general population [17–20, 22, 23, 25, 27, 28, 32, 34, 37–40], three studies reported CKDu prevalence [24, 28, 29, 31] in the general population and Two studies [28, 36] reported prevalence of both CKD and CKDu in the general population. Two studies were conducted in the diabetes population [26, 30] and one study [21] in the hypertensive population.

## PRISMA Flow Diagram

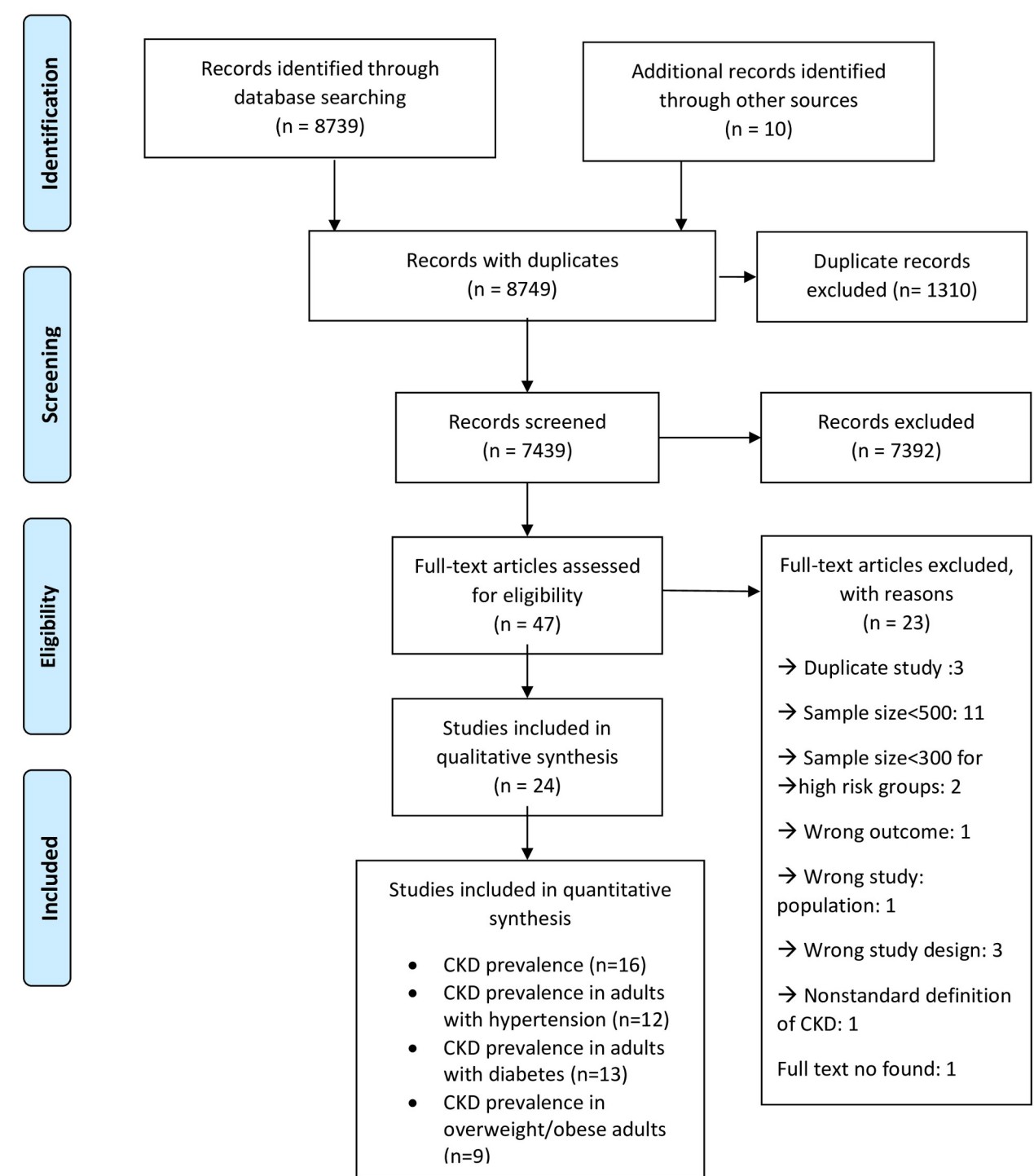

**Fig 1. Flow diagram of the study selection.**

The characteristics of the included studies are summarised in Table 1. Most studies were from India (n = 13) [17–19, 27–30, 34–36, 38, 39] and rest were from Bangladesh (n = 4) [20, 22, 23, 26], Nepal (n = 3) [32, 33, 40], Sri Lanka (n = 2) [24, 31], Pakistan (n = 1) [25]. One study reported data from Bangladesh, Sri Lanka and Pakistan [21]. However, both the studies from Sri Lanka reported only the prevalence of CKDu (12, 19). Altogether 16 studies were conducted in community settings [17–19, 21, 23–25, 28, 29, 31–36, 40] and the remaining were hospital- or clinic-based studies. Out of 21 studies that reported CKD prevalence, 16 studies used single equation to estimate GFR: MDRD: 11 studies [20, 26–28, 30, 32, 33, 36, 38, 40], CKD-EPI: 4 studies [17, 19, 21, 25], CG: one study [37]. Five studies reported GFR using two equations (MDRD and CG: three studies [18, 22, 23], MDRD and CKD-EPI: two studies [34, 39]. Most studies reported using Jaffe's kinetic assay for creatinine assessment [17–19, 22, 23, 29, 34, 35, 37–40] (n = 12), while seven studies [20, 26, 27, 30, 32, 33] did not report assessment method for creatinine. Only four studies followed up participants at 3 months [22, 23, 26, 40]. Only seven studies reported calibration of creatinine measurement by isotope dilution mass spectrometry assays [17, 21, 25, 29, 31, 34, 37] which reduces variability in measured serum creatinine values [41]. Most commonly used method for measurement of proteinuria is urinary dipstick [18–20, 22, 23, 27, 28, 30, 32–35, 38, 39], however three studies did not report measurement method for proteinuria [26, 36, 37].

## Assessment of risk of bias

Thirteen studies [20, 22–24, 26, 27, 30, 32–34, 37–39] scored 2 or less out of 4 possible points in the external validity domain whereas twenty-two studies [17–22, 24–26, 28–33, 35–40] scored all 6 points and the remaining two studies [23, 34] scored 5 points in the internal validity domain. Overall, ten studies [17–19, 21, 25, 28, 29, 31, 36, 40] were judged to be of high methodological quality and the remaining 14 studies [20, 22–24, 26, 27, 30, 32–35, 37–39] were judged to be of moderate methodological quality (S2 Table).

## Prevalence of CKD and CKDu in the general population

The overall prevalence of CKD was 14% (95% CI 11–18%, n = 50,494, 16 studies, Fig 2) in the general population, 15% (95% CI 11–20%, n = 22, 973, 15 studies, S1 Fig) in adult males and 13% (95% CI 10–17%, n = 24,528, 15 studies, S2 Fig) in adult females. The prevalence of CKD in general population was 16% (95% CI 12–21%) in India, 14% (95% CI 12–17%) in Bangladesh, 12% (95% CI 11–14%) in Pakistan and 6% (95% CI 6–7%) in Nepal (Fig 2). One study [33] did not report estimated CKD prevalence therefore could not be pooled in the meta-analysis. It reported a 14.4% prevalence of low GFR (eGFR<60 mL/min/1.73 m2) and 7% prevalence of proteinuria in Nepal. Similarly, another study [35] from India which could not be pooled in a meta-analysis reported 13.3% prevalence of low GFR by CG equation and 4.2% prevalence of low GFR by MDRD equation. The same study [35] also reported a 2.2% prevalence of proteinuria in the Indian population. The prevalence of GFR $< 60$ mL/min/1.73 $m^2$ was 6% (95% CI 4–9%, n = 64,143, 18 studies, S3 Fig). In subgroup analysis by GFR estimation equation, CKD prevalence was 13% (95% CI 10–17%) by MDRD, 15% (95% CI 10–21%) by CKD EPI and 18% (95% CI 15–21%) by CG (S4–S6 Figs). Those studies that followed up participant at 3 months reported a slightly lower CKD prevalence of 12% (95% CI 5–20%) (S7 Fig). Similarly subgroup analysis also revealed that a higher prevalence of CKD in population of age group 40 years and older compared to age group below 40 years (18% vs. 6%, S8 Fig). The prevalence of CKDu in the endemic population was 8% (95% CI 3–16%, 5 studies) (S9 Fig).

**Table 1. Detailed characteristics of the included studies assessing the prevalence of CKD/CKDu in South Asia.**

| Author (ref) year of study | Country | Sample size (N), sampling technique | Population characteristics | Diagnostic method | | CKD Prevalence | CKD prevalence in high-risk groups (Diabetes/ HTN/ Obese) |
|---|---|---|---|---|---|---|---|
| | | | | Type of creatinine assay, Creatinine: Equation, 3 months follow up measurement (yes/ No) | Proteinuria measurement | | |
| **Anand 2015 [17]** | India | 9797, Multistage cluster sampling from two Indian cities Delhi and Chennai (Cardiometabolic Risk Reduction in South Asia surveillance Study) | Setting: community, urban, mean age: 41.4 (±12.7) years for Chennai; 44.4 (±13.9) years for Delhi: 56% women in Chennai, 51%women in Delhi | Jaffe method with isotope dilution mass spectrometry (IDMS)-traceable assays<br><br>Equation: CKD-EPI<br><br>Follow up: No | Immuno-turbidimetric assay | Overall:7.5%;:8.7% (after age standardization)<br><br>Male: 7.3%<br><br>Female: 7.7% | Diabetic:15.4%, hypertensive:7%, Obese:9.6% |
| **Anupama 2014 [18]** | India | 2091, Random sampling in the villages of Hosakoppa, Indiranagar and part of Gajanur area in Shimoga District | Setting: Community, rural, mean age: 39.3, mean BMI: NR, 54.4% females | Jaffe's method<br><br>Equation: MDRD, CG<br><br>Follow up: No | Dipstick | Overall: 6.3% (MDRD), 16.69% (CG)<br><br>Male: 8.1%<br><br>Female: 4.7% | NR |
| **Farag 2020 [19]** | India | 1184, systematic random sampling from eight administrative division in Andhra Pradesh (Screening and Early Evaluation of Kidney disease–SEEK- Andhra) | Setting: mixed, community, mean age: 44.6 (± 14), mean BMI: 21.8, Hypertension: 45.5%, diabetes: 22.1% | Jaffe's method<br><br>Equation: CKD-EPI<br><br>Follow up: No | Dipstick | Overall: 32.2%,<br><br>Male: 37.8%,<br><br>female: 27.7% | Diabetes: 33.2% Hypertension: 36.4% Overweight/obese: 22.1% |
| **Fatema 2013 [20]** | Bangladesh | 634, Consecutive sampling in Dhaka city | Setting: Screening program, urban, mean age: 37, mean BMI: 23.7 (±3.4), 21.7% females, hypertension: 25.2%, diabetes: 8.5% | NR<br><br>Equation: MDRD<br><br>Follow up: No | Dipstick | Overall: 12.8% | Overweight/obese: 13.8%<br><br>Hypertension: 15.6% |
| **Feng 2019 [21]** | Bangladesh, Pakistan and Sri Lanka | 2349, Stratified cluster random sampling of hypertensive adults | Settings: Community, mixed, mean age:58.8, mean BMI: 24.7 (±5), 64.5% female, diabetes: 26.7% | Calibrated for IDMS<br><br>Equation: CKD-EPI<br><br>Follow up: No | Nephelometry using the array systems method | Overall:38.1% (Sri Lanka = 58.3; Bangladesh = 36.4%; Pakistan = 16.9%) | |
| **Hasan 2012 [22]** | Bangladesh | 1240, Consecutive sampling among outpatients in Nephrology department of Mymensingh medical college | Settings: Outpatient, mixed, mean age: 37.1 (± 10.9), 48% female | Kinetic method<br><br>Equation: MDRD, CG<br><br>Follow up: yes | Dipstick | Overall: 19.5% (MDRD), 19% (CG)<br><br>Male: 22.8%<br><br>Female: 15.9% | NR |
| **Huda 2012 [23]** | Bangladesh | 1000, multistage clustered sampling in certain selected slum areas of Mirpur at Dhaka city | Settings: Urban, community, mean age:34.39 (±12.70), mean BMI:NR, 66.6% females. | Jaffe kinetic assay not standardized by IDMS<br><br>Equation: MDRD, CG<br><br>Follow up: yes | Dipstick | Overall: 13.1% (MDRD), 16% (CG)<br><br>Male: 14.6%,<br><br>Female: 12.3% | Diabetes: 10.6%, hypertension: 31.9% |
| **Jessani 2014 [25]** | Pakistan | 2873, Cluster random sampling in 12 representative communities in Karachi | Setting: Urban, community, mean age: 51.5(±10.7), mean BMI: 25.8 (±5.5), 52.2% females, hypertension: 44.9%, diabetes: 21.4% | Roche enzymatic creatinine assay with IDMS<br><br>Equation: CKD-EPI<br><br>Follow up: No | Nephelometry by the Array Systems method | Overall: 12.5%<br><br>Female: 13.34%<br><br>Male: 11.57% | Hypertension: 20%, Diabetes: 24.4% |
| **Khanam 2016 [26]** | Bangladesh | 1317, Institute of Research and Rehabilitation in Diabetes, Endocrine and Metabolic Disorders (BIRDEM) registry | Setting: mixed; Hospital records of diabetes patients, 45.3% women | NR<br><br>Equation: MDRD<br><br>Follow up: yes | NR | Overall: 13.9%<br><br>Female:21.3%<br><br>Male:7.8% | (Note: prevalence of CKD was 7.1% based creatinine level > 1.2 mg/dl) |
| **Mahapatra 2016 [27]** | India | 1104, Convenience sample of employees from the six Central Government Offices | Setting: Outpatient, mean age: 43.5 (±9.5), mean BMI:25.3 (±4.02), 38.6% females. | NR<br><br>Equation: MDRD<br><br>Follow up: no | Dipstick | Overall: 27.7% | Hypertension: 30.6%, diabetes: 41.3% |

*(Continued)*

**Table 1.** (Continued)

| Author (ref) year of study | Country | Sample size (N), sampling technique | Population characteristics | Diagnostic method | | CKD Prevalence | CKD prevalence in high-risk groups (Diabetes/ HTN/ Obese) |
|---|---|---|---|---|---|---|---|
| | | | | **Type of creatinine assay, Creatinine: Equation, 3 months follow up measurement (yes/ No)** | **Proteinuria measurement** | | |
| **Mohanty 2020 [28]** | India | 2978, Multistage cluster sampling in the Narsinghpur block of Cuttack, Odisha | Setting: Community, Rural, Mean age: NR,62.7% female | Enzymatic method | Dipstick | Overall:14.3%; | 3.5% had CKD with either diabetes or hypertension |
| | | | | Equation: MDRD | | Female:53% | |
| | | | | | | Male:47% | |
| | | | | Follow up: no | | CKDu:11% | |
| **Rajput 2017 [30]** | India | 2866, Consecutive sampling | Setting: Outpatient diabetes patients from multiple centers, Mean age: 53.4(±11.9), mean BMI: 27.3 (±4.8), 46% female | NR | Dipstick | | Diabetes: 48.4% |
| | | | | Equation: MDRD | | | |
| | | | | Follow up: no | | | |
| **Selected NCDs 2019 [40]** | Nepal | 11,475, nationwide Multistage cluster sampling | Setting: mixed, Community, mean age: NR, 61.7% females, BMI: NR, hypertension: 36.4%, diabetes: 8.23% | Jaffe's kinetic assay | Immunoturbidimetric tests | Overall: 6.0% | Diabetes: 19.7%, hypertension: 10.7%, overweight/obese: 6.98% |
| | | | | Equation: MDRD | | Male: 6.57% | |
| | | | | Follow up: yes | | Female: 5.62% | |
| **Sharma 2013 [32]** | Nepal | Subset of 1000 participants was used in the analysis, Consecutive sampling 4 VDS out of 19 VDCs in Dharan municipality | Setting: Urban, community, mean age: 42.9 (±14.9), mean BMI:NR, 52.1% females. | NR | Dipstick | Overall: 10.6% | NR |
| | | | | Equation: MDRD | | | |
| | | | | Follow up: no | | | |
| **Sharma 2010 [33]** | Nepal | 8398, Purposive sampling in kidney disease screening program in Eastern Nepal | Setting: mixed, Community, mean age:38, mean BMI:22.8, 62% females. | NR | Dipstick | NR, reported participants with reduced GFR | NR |
| | | | | Equation: MDRD | | | |
| | | | | Follow up: no | | | |
| **Singh 2009 [35]** | India | 5252, Multistage cluster sampling in Delhi and adjoining regions. | Setting: mixed, Community, mean age:38.9, mean BMI:23.2, 39.9% females. | Jaffe kinetic assay | Dipstick | NR, reported participants with reduced GFR | NR |
| | | | | Equation: MDRD | | | |
| | | | | Follow up: no | | | |
| **Singh 2013 [34]** | India | 5588, Consecutive sampling in 13 medical centers (Screening and Early Evaluation of Kidney disease–SEEK) | Setting: Urban, Community, mean age: 45.22 (±15.2), mean BMI: NR, 44.9% females. | Jaffe Colorimetric method with IDMS-traceable assays | Dipstick | Overall: 17.2% (MDRD), 16.4% (CKD-EPI) | Diabetes: 28.9%, Hypertension: 25.8% |
| | | | | Equation: MDRD, CKD-EPI) | | Male: 15.5% | |
| | | | | Follow up: no | | Female: 19.8% | |
| **Tatapudi 2019 [36]** | India | 2210, multistage sampling in the southern Indian state of Andhra Pradesh | Setting: mixed, Community, mean age: 43.2 (±14.2), mean BMI: NR, 55.7% women, hypertension: 16.7%, diabetes: 7.2% | NR | NR | Overall: 18.2%, | Diabetes: 37.1%, Hypertension: 36% |
| | | | | Equation: MDRD | | Male: 19.1% | |
| | | | | | | Female: 17.5% | |
| | | | | Follow up: no | | CKDu: 13.3% | |
| **Trivedi 2016 [37]** | India | 2350, Purposive sampling in kidney disease screening program in Gujrat | Setting: Semi urban, Outpatient, mean age: 48.16 (±14), mean BMI: 25.62 (±5.25), 38.8% females | Jaffe Colorimetric method with IDMS assays | NR | Overall: 20.93% | NR |
| | | | | | | Male: 21% | |
| | | | | Equation: CG | | Female: 20.8% | |
| | | | | Follow up: no | | | |
| **Varma 2010 [39]** | India | 3398, Purposive sampling among central government employees | Setting: outpatient, mean age: 35.64 ± 8.72, mean BMI: NR, 34% females, hypertension: 13.15%, diabetes: 1.53% | Jaffe's kinetic assay | Dipstick | Overall: 15% (MDRD) & 13.1% (CKD-EPI) | NR |
| | | | | Equation: MDRD & CKD EPI | | | |
| | | | | Follow up: no | | Male: 13.04% (MDRD) & 12.62% (CKD-EPI) | |
| | | | | | | Female: 19.13% (MDRD) & 14.13 (CKD-EPI) | |

(*Continued*)

**Table 1.** (Continued)

| Author (ref) year of study | Country | Sample size (N), sampling technique | Population characteristics | Diagnostic method | | CKD Prevalence | CKD prevalence in high-risk groups (Diabetes/ HTN/ Obese) |
|---|---|---|---|---|---|---|---|
| | | | | Type of creatinine assay, Creatinine: Equation, 3 months follow up measurement (yes/ No) | Proteinuria measurement | | |
| **Varma 2011** [38] | India | 1572, purposive sampling; all the army personnel posted in the station were screened | Setting: Outpatient, mean age: 34.72 (±7.57), mean BMI: NR, hypertension: 8.49%, diabetes 1.09% | Jaffe's kinetic assay<br>Equation: MDRD<br>Follow up: no | Dipstick | Overall:9.54% | NR |
| **Chronic Kidney Disease of Unknown Origin** | | | | | | | |
| **Jayalatike 2013** [24] | Sri Lanka | 4777, cluster random sampling | Setting: rural, Community, mean age: NR, mean BMI: NR | persistent albuminuria, i.e. albumin creatinine ratio (ACR)≥ 30 mg/g in an initial urine sample and at a repeat visit, no past history of glomerulonephritis, pyelonephritis, renal calculi or snake bite, not on treatment for diabetes | | Overall CKDu prevalence: 15.3% | |
| **Ruwanpathirana 2019** [31] | Sri Lanka | 4803, cluster sampling | Setting: mixed, community, mean age: NR, mean BMI: NR | a single measure of impaired kidney function (eGFR< 60 mL/min/1.7m$^2$, using the CKD-Epi formula) in the absence of hypertension, diabetes or heavy proteinuria | | Overall CKDu prevalence: 6% | |
| **O'callaghan 2019** [29] | India | 12500, three population-based studies, CAARS, UDAY and ICMR-CHD studies | Setting: mixed, community, mean age: NR, mean BMI: NR | CKD in individuals with known risk factors for CKD (ie, diabetes and hypertension) or evidence of primary glomerular disease (as assessed by heavy proteinuria). | | Overall CKDu prevalence: 2% | |

#NR: Not reported, CKD: Chronic Kidney Disease, CKDu: Chronic Kidney Disease of Unknown Origin, MDRD: Modification of Diet in Renal Disease, CKD-EPI: Chronic Kidney Disease Epidemiology Collaboration, CG: Cockcroft-Gault formula.

We found substantial heterogeneity across the included studies in the pooled estimates for overall CKD prevalence, prevalence disaggregated by age groups, sex, across eGFR estimating equations and CKDu prevalence. In the meta-regression, the regression coefficient for the mean age, mean BMI and survey year was statistically non-significant (S3 Table). Publication bias was detected across studies reporting on the prevalence of CKD in the general population, with the asymmetrical funnel plot and significant Egger test (S10 Fig; p-value 0.002).

### Prevalence of CKD in high-risk populations

The prevalence of CKD was 27% (95% CI 20–35%, 12 studies) in adults with hypertension, 31% (95% CI 22–41%, 13 studies) in adults with diabetes and 14% (95% CI 10–19%, 9 studies) in adults who were overweight/obese (Figs 3–5). We found substantial heterogeneity across the included studies in the pooled estimates for CKD prevalence in high risk populations. We found no evidence of publication bias across studies reporting on the prevalence of CKD in the adults with hypertension and the adults with diabetes, with the symmetrical funnel plot and non-significant Egger test (S11 and S12 Figs).

### Discussion

Our study found that prevalence of CKD was nearly 14% in general population and among those who were overweight/obese. CKD prevalence was two times higher among adults with hypertension and diabetes compared to general population. These findings are comparable to CKD prevalence in African continent (15.8%, 95% CI 12.1–19.9%) [42] and global prevalence

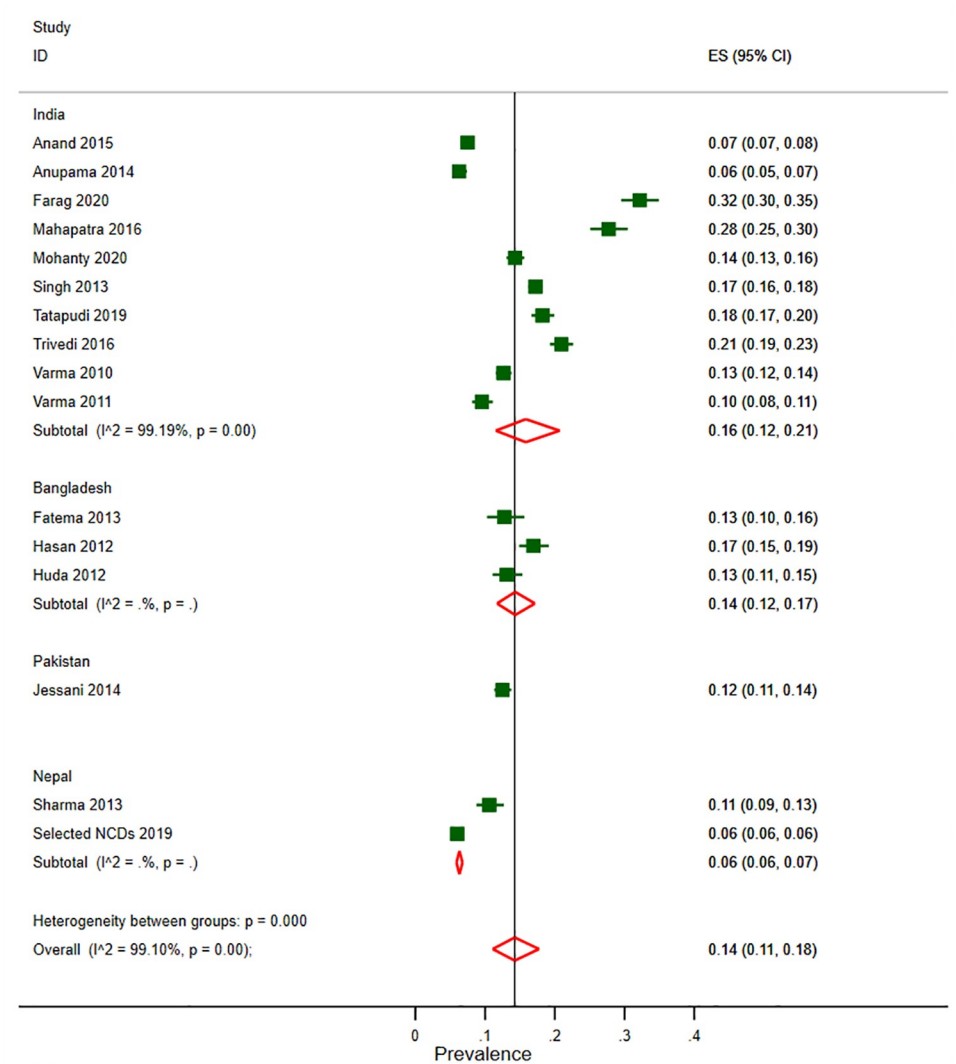

**Fig 2. Prevalence of CKD in general populations of adults in South Asia.** Black boxes represent the effect estimates (prevalence) and the horizontal bars are for the 95% confidence intervals (CIs). The diamond is for the pooled effect estimate and 95% CI and the dotted vertical line centered on the diamond has been added to assist visual interpretation.

of CKD (13.4%, 95% CI 11.7–15.1%) [43]. Our pooled estimate in a high-risk population (adults with hypertension and adults with diabetes) is comparable to the prevalence of CKD in high risk population in the African continent [42]. There was high heterogeneity in the pooled estimates of CKD across the studies for the general population as well as in the high risk population. In subgroup analysis based on the three main equations used to estimate the GFR yielded different prevalence estimates. The prevalence estimates obtained from the Cockcroft formula was higher than the prevalence obtained using MDRD or CKD-EPI equations. While majority of included studies used Jaffe kinetic method, others lacked information on serum creatinine assessment. Therefore, we could not assess whether the variation in laboratory assessment of serum creatinine contributed to heterogeneity in pooled estimates. We noted variation in the burden of CKD by countries of the South Asia region: 16% in India to 6% in Nepal. Furthermore, the burden of CKD also varied widely within the countries itself, ranging

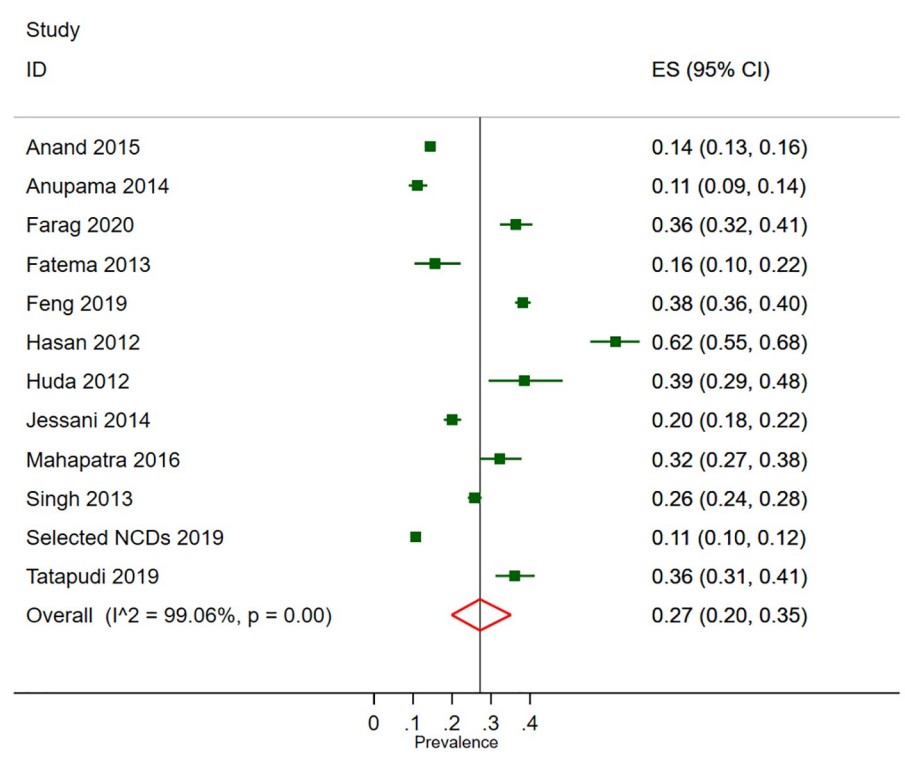

**Fig 3. Prevalence of CKD in adults with hypertension in South Asia.**

from 6 to 32% in India, 13 to 17% in Bangladesh and 6 to 11% in Nepal. This might be because of diversity in geographic, socio-economic status, lifestyle and culture across South Asia [44]. The global burden of disease 2019 also suggests a wide variation in CKD prevalence in south Asian countries ranging from 4.5 to 13.5% [45] (S13 Fig). CKD was more prevalent in men (15%) than in women (13%). This variance could be explained by higher muscle mass in men than women in general as serum creatinine concentration in an individual depends on his/her muscle mass. These findings corroborate with the existing literature on gender differences in CKD prevalence [46, 47]. In subgroup analysis, prevalence of CKD was found three times higher in individuals aged 40 years and older compared to their younger counterparts. This might be because of gradual decline in renal function (GFR) at the rate of 0.75 to 1 ml/min/ year after the age of 40 years [48]. Additionally, the higher prevalence of cardiometabolic risk factors like diabetes and hypertension in older age groups [49, 50] also explains the higher prevalence of CKD in in age group 40 years and older.

Metabolic risk factors, such as diabetes, hypertension, and obesity, have been described as strong risk factors of both ESRD and CVD events in large observational studies [6, 51]. Interestingly, countries, with a lower prevalence of hypertension reported a lower prevalence of CKD [49] and likewise modestly higher prevalence of CKD in India [52] and China [53] coincided with a higher prevalence of hypertension. Considering the rise in metabolic risk factors such as diabetes, hypertension, and obesity in the South Asian continent [49, 50, 54–56], this might further increase the CKD burden in these countries in coming years. On the other hand, a significant proportion of deaths in CKD patients (around 40%) occur prematurely

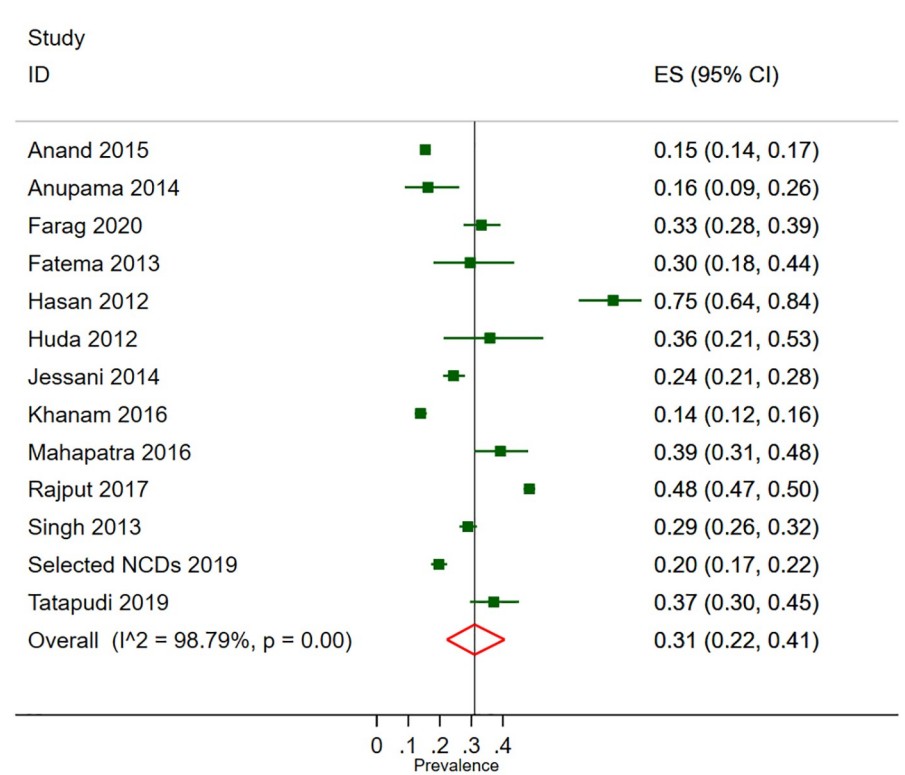

**Fig 4. Prevalence of CKD in adults with diabetes in South Asia.**

(before age 65) primarily due to cardiovascular complications [57, 58]. Therefore, these premature deaths can be prevented through strategies to slow the progression of CKD and optimal control of cardiometabolic risk factors.

On one hand, extrapolating the 14% prevalence of CKD in this study to South Asia's 1.3 billion adult population (aged ≥20 years), we estimate that 143–234 million are at risk of some forms of kidney damage. This could mean nearly 22,880 to 37,440 would require treatment for ESRD every year [59]. However, the shortage of renal replacement services in South Asian countries is concerning. A study reported that there were only 900 Nephrologist and 5,500 dialysis centers in India which provide dialysis services to an estimated 55,000 ESRD patients [60]. The situation in other South Asian countries is even more disheartening such as in Nepal. There are only 36 Nephrologists and 42 dialysis centers that provide dialysis services to an estimated 2000 ESRD patients [61, 62]. Recent anecdotes also showed an increase in waiting time for patients with ESRD in dialysis centers and increasing demand for kidney transplantation for ESRD patients in South Asia [8]. Therefore, providing universal nephrology services remained an important priority in South Asia. Targeting risk factors for CKD, specifically blood pressure, blood glucose, and lipids, through a combination of active lifestyle intervention and pharmacotherapy can lead to better prognosis among those with kidney damage [63]. These interventions can be implemented at minimal cost and have been shown to decrease the burden of ESRD, risk of cardiovascular complications as well as morbidity and mortality from non communicable diseases (NCDs) [64].

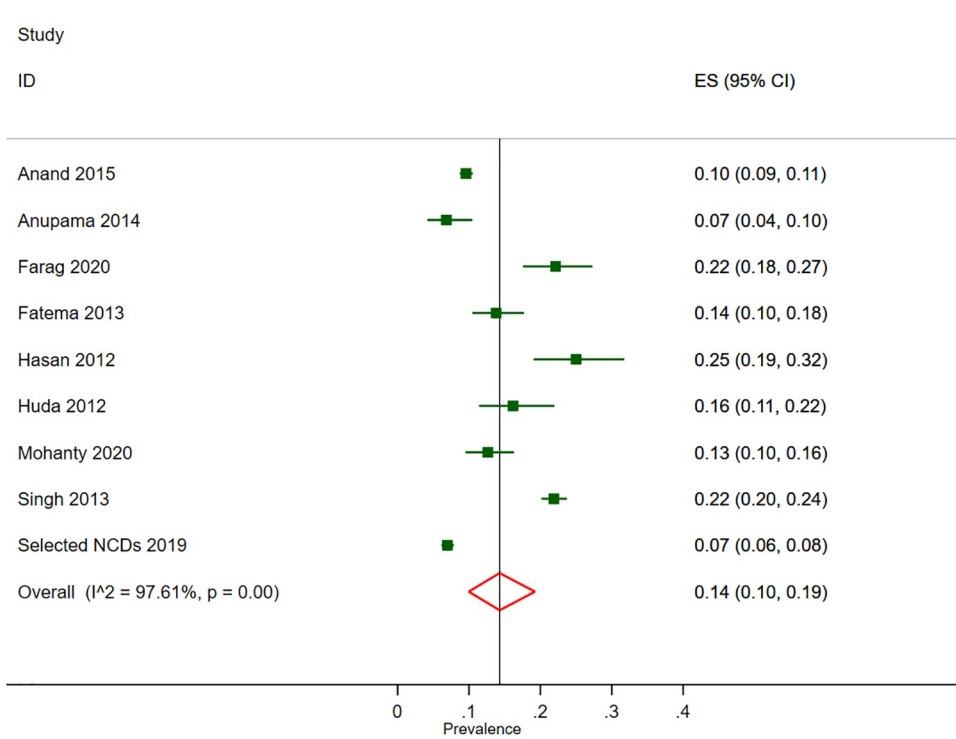

**Fig 5. Prevalence of CKD in overweight/obese adults in South Asia.**

In some regions in South Asia like the Indian states of Andhra Pradesh and Odisha [29] and the Anuradhapura district of Sri Lanka [31], chronic kidney disease in agricultural population without established risk factors (such as hypertension, diabetes) was identified and termed as CKDu. The studies that reported prevalence of CKDu identified in this review were from these endemic regions and therefore cannot be generalised to the entire population of the respective country. The pooled prevalence of CKDu in the endemic regions of South Asia was 8% (95% CI 3–16%).

The major strength of this study is an extensive search of literature in multiple electronic databases and reference list of identified studies. However, we might have missed some publications in the local language, including reports from national authorities/registries. The review was conducted strictly adhering to MOOSE guidelines and each was performed independently by two reviewers to minimise bias. The findings of this study, however, need to be interpreted with caution. There was substantial heterogeneity in the prevalence estimates across the studies, which could not be explained by subgroup analyses and meta-regression. The sources of these heterogeneities may stem from differences in characteristics of the study population and analytical variability in the measurement of serum creatinine [65, 66] and urinary albumin [67]. Moreover, most studies assessed urinary protein with dipstick instead of albumin which correlates poorly with albumin to creatinine ratio [68]. Most studies did not conduct the follow up at 3 months to demonstrate chronicity thus failed to detect the cases with transient changes in GFR and albuminuria [69]. We could not perform subgroup analysis for other covariates that may have contributed to heterogeneity across the included studies, such as age, residence.

## Conclusions

This study highlighted CKD as an important health priority in South Asia. One in every seven participants in our study had CKD. CKD was nearly twice prevalent in adults with hypertension and diabetes compared to the general population. This shows an urgency for lifestyle intervention to target common NCD risk factors to reduce the progression of CKD and future CVD events.

## Supporting information

**S1 Checklist. PRISMA 2009 checklist.**
(PDF)

**S1 Table. Detailed search strategy for assessing the pooled prevalence of CKD/CKDu in South Asia.**
(PDF)

**S2 Table. Quality assessment of the included studies using the Hoy et al. risk of bias assessment tool.**
(PDF)

**S3 Table. Meta-regression analysis for the variation of the prevalence of CKD in the general population of adults in South Asia.**
(PDF)

**S1 Fig. Prevalence of CKD in adult males in South Asia.**
(TIF)

**S2 Fig. Prevalence of CKD in adult females in South Asia.**
(TIF)

**S3 Fig. Prevalence of GFR < 60 mL/min/1.73 m2 in the general population of adults in South Asia.**
(TIF)

**S4 Fig. Prevalence of CKD in the general population of adults in South Asia based on MDRD equation for GFR estimation.**
(TIF)

**S5 Fig. Prevalence of CKD in the general population of adults in South Asia based on CKD-EPI equation for GFR estimation.**
(TIF)

**S6 Fig. Prevalence of CKD in the general population of adults in South Asia based on Cockcroft-Gault equation for GFR estimation.**
(TIF)

**S7 Fig. Prevalence of CKD in general population of adults in South Asia (studies with 3 months follow up).**
(TIF)

**S8 Fig. Prevalence of CKD in age group <40 years and age group $\geq$ 40 years.**
(TIF)

**S9 Fig. Prevalence of CKDu in endemic populations of adults in South Asia.**
(TIF)

**S10 Fig. Egger's test and funnel plot for publication bias for overall CKD prevalence.**
(TIF)

**S11 Fig. Egger's test and funnel plot for publication bias for prevalence of CKD in adults with hypertension.**
(TIF)

**S12 Fig. Egger's test and funnel plot for publication bias for prevalence of CKD in adults with diabetes.**
(TIF)

**S13 Fig. Forrest plot showing prevalence of chronic kidney disease in South Asian countries from global burden of disease 2019.**
(TIF)

## Author Contributions

**Conceptualization:** Nipun Shrestha, Shiva Raj Mishra.

**Data curation:** Nipun Shrestha, Sanju Gautam, Raja Ram Dhungana.

**Formal analysis:** Nipun Shrestha.

**Investigation:** Nipun Shrestha.

**Methodology:** Nipun Shrestha, Shiva Raj Mishra, Raja Ram Dhungana.

**Visualization:** Shiva Raj Mishra.

**Writing – original draft:** Nipun Shrestha.

**Writing – review & editing:** Nipun Shrestha, Sanju Gautam, Shiva Raj Mishra, Salim S. Virani, Raja Ram Dhungana.

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
