## [Decision Letter · Decision Letter 0]

22 Jul 2021

PONE-D-21-16355

Burden of chronic kidney disease in the general population and high-risk groups in South Asia: a systematic review and meta-analysis

PLOS ONE

Dear Dr. Shrestha,

Thank you for submitting your manuscript to PLOS ONE. After careful consideration, we feel that it has merit but does not fully meet PLOS ONE’s publication criteria as it currently stands. Therefore, we invite you to submit a revised version of the manuscript that addresses the points raised during the review process.

As the Academic Editor I congratulate you with the work performed and the prepared manuscript. Please address the comments made by the reviewers, and also the following items:

1. Please indicate why your search strategy did not retrieve the following articles containing data about CKD prevalence in Pakistan:

- Imran S, Sheikh A, Saeed Z, Khan SA, Malik AO, Patel J, Kashif W, Hussain A. Burden of chronic kidney disease in an urban city of Pakistan, a cross-sectional study. J Pak Med Assoc. 2015; 65(4): 366-9.

- Alam A, Amanullah F, Baig-Ansari N, Lotia-Farrukh I, Khan FS. Prevalence and risk factors of kidney disease in urban Karachi: baseline findings from a community cohort study. BMC Res Notes. 2014; 7: 179.

2. Please compare the CKD prevalence indicated in the articles included in the meta-analysis for each country, and also the summary CKD prevalence for the South Asia with the estimates reported by the GBD Study (GBD Chronic Kidney Disease Collaboration. Global, regional, and national burden of chronic kidney disease, 1990–2017: a systematic analysis for the Global Burden of Disease Study 2017. Lancet. 2020; 395: 709-733. doi: 10.1016/S0140-6736(20)30045-3)

We look forward to receiving your revised manuscript.

Kind regards,

Boris Bikbov

Academic Editor

PLOS ONE

Journal Requirements:

"The funders had no role in study design, data collection and analysis, decision to

publish, or preparation of the manuscript."

c) If you did not receive any funding for this study, please state: “The authors received no specific funding for this work.”

"Nipun Shrestha, Sanju Gautam, Shiva Raj Mishra and Raja Dhungana declare that no

competing interests exists.

Salim Virani: Research funding: US Department of Veterans Affairs, World Heart

Federation, Tahir and Jooma Family Honorarium: American College of Cardiology

(Associate Editor for Innovations, acc.org)"

Reviewers' comments:

Reviewer's Responses to Questions

**Comments to the Author**

1. Is the manuscript technically sound, and do the data support the conclusions?

Reviewer #1: Yes

Reviewer #2: Yes

2. Has the statistical analysis been performed appropriately and rigorously? 

Reviewer #1: Yes

Reviewer #2: Yes

3. Have the authors made all data underlying the findings in their manuscript fully available?

Reviewer #1: No

Reviewer #2: Yes

4. Is the manuscript presented in an intelligible fashion and written in standard English?

Reviewer #1: Yes

Reviewer #2: Yes

5. Review Comments to the Author

Reviewer #1: Shrestha N et al. Burden of chronic kidney disease in the general population and high-risk groups in South Asia: a systematic review and meta-analysis.

The method of this systematic review was adequately described and considered appropriate, following the MOOSE guideline. There some questions and suggestions.

- Are there any language restrictions for the eligible studies? Some publications in local language including report from national authorities/registries, if presence, may be missed.

- Sensitivity analysis regarding risk of bias should be done as a part of the investigations of heterogeneity.

- If data available, the result should be separated as difference stages of CKD by GFR with or without albuminuria. The prevalence of advanced stage of CKD might better show the burden of disease. Prevalence of CKD by GFR may be reported separately from proteinuria due to some limitation of testing in each study.

- Most of the included studies sampled population from specific area, such as some province or city. How the authors ensure the representativeness for the overall population in South Asia?

Reviewer #2: This is a comprehensive review of CKD in S East Asia.

Minor comments:

Please state the start and end dates of the systematic review

Table 1: Feng et al Move overall prevalence to appropriate column

A limitation is that there is no indication if investigators of prevalence data used IDMS traceable creatinine

6. PLOS authors have the option to publish the peer review history of their article (what does this mean?). If published, this will include your full peer review and any attached files.

Reviewer #1: No

Reviewer #2: **Yes: **Jaya A George

---

## [Author Response · Author response to Decision Letter 0]

11 Aug 2021

Editor’s comment

1. Please indicate why your search strategy did not retrieve the following articles containing data about CKD prevalence in Pakistan:

- Imran S, Sheikh A, Saeed Z, Khan SA, Malik AO, Patel J, Kashif W, Hussain A. Burden of chronic kidney disease in an urban city of Pakistan, a cross-sectional study. J Pak Med Assoc. 2015; 65(4): 366-9.

- Alam A, Amanullah F, Baig-Ansari N, Lotia-Farrukh I, Khan FS. Prevalence and risk factors of kidney disease in urban Karachi: baseline findings from a community cohort study. BMC Res Notes. 2014; 7: 179.

Author’s response: These publications were identified in our searches, however, were excluded because the sample size was less than 500. This has been clearly stated under the heading exclusion criteria. 

2. Please compare the CKD prevalence indicated in the articles included in the meta-analysis for each country, and also the summary CKD prevalence for the South Asia with the estimates reported by the GBD Study (GBD Chronic Kidney Disease Collaboration. Global, regional, and national burden of chronic kidney disease, 1990–2017: a systematic analysis for the Global Burden of Disease Study 2017. Lancet. 2020; 395: 709-733. doi: 10.1016/S0140-6736(20)30045-3)

Author’s response: Thank you for the comments. We have now added the supplementary forest plot with prevalence estimates of chronic kidney disease from all the countries from Global Burden of Disease 2019 – Supplementary figure 13. We also added following sentences in discussion comparing our estimates with global burden of disease 2019 estimates (lines 277-78, page 12).

"The global burden of disease 2019 also suggest a wide variation in CKD prevalence in south Asian countries ranging from 4.5 to 13.5% (Supplementary figure 13)”.

Reviewer #1: 

3. Shrestha N et al. Burden of chronic kidney disease in the general population and high-risk groups in South Asia: a systematic review and meta-analysis.

The method of this systematic review was adequately described and considered appropriate, following the MOOSE guideline. There are some questions and suggestions.

- Are there any language restrictions for the eligible studies? Some publications in local language including report from national authorities/registries, if presence, may be missed.

Author’s response: There were no language restrictions for the eligible studies. However, we do agree that some publications in local language including report from national authorities/registries might have been missed. We have added following statement to acknowledge this limitation (lines 325-27, page 14). 

“The major strength of this study is an extensive search of literature in multiple electronic databases and reference list of identified studies. However, we might have missed some publications in local language including report from national authorities/registries”.

4. Sensitivity analysis regarding risk of bias should be done as a part of the investigations of heterogeneity.

Author’s response: Thank you for your comments. We judged 10 studies at low risk of bias and 14 studies at moderate risk of bias. Since there were no studies judged at high risk of bias, we did not perform sensitivity analysis excluding studies at high risk of bas.

5. If data available, the result should be separated as difference stages of CKD by GFR with or without albuminuria. The prevalence of advanced stage of CKD might better show the burden of disease. Prevalence of CKD by GFR may be reported separately from proteinuria due to some limitation of testing in each study.

Author’s response: We agree that prevalence of advanced stage of CKD might better show the burden of disease. Prevalence of CKD by GFR stages were not performed because most of the studies had small sample sizes. We have added a subgroup analysis of CKD by GFR < 60 mL/min/1.73 m2 which corresponds to CKD Stage 3-5 by GFR with or without albuminuria (Supplementary figure 3). We have also added following sentence in results section (lines 228-29, page 10).

“The prevalence of GFR < 60 mL/min/1.73 m2 was 6% (95% CI 4- 9%, n = 64,143, 18 studies, Supplementary figure 3)”.

6. Most of the included studies sampled population from specific area, such as some province or city. How the authors ensure the representativeness for the overall population in South Asia?

Author’s response: Like many systematic reviews and metanalysis in this topic, our study is limited by geographical coverage of individual studies. Therefore, we have made careful consideration not to generalise the findings at the population level in South Asia both in the results and discussion section of the paper. Also, being mindful that it would be logistically challenging to build individual studies that capture the whole population - unless such measurements (of kidney function) are routinely done and can be captured through central databases/repositories (e.g. NHS).

Reviewer #2: 

This is a comprehensive review of CKD in S East Asia.

Minor comments:

7. Please state the start and end dates of the systematic review

 Author’s response: This information has already been incorporated into the search strategy of the manuscript. Please refer to the “Search strategy and selection criteria” section (lines 87, page 4).

8. Table 1: Feng et al Move overall prevalence to appropriate column

Author response: Thank you for identifying this typo. We have not moved overall prevalence to appropriate column.

9. A limitation is that there is no indication if investigators of prevalence data used IDMS traceable creatinine.

Author response: We have now added following statement on studies that used IDMS traceable creatinine.

“Only seven studies reported calibration of creatinine measurement by isotope dilution mass spectrometry assays which reduces variability in measured serum creatinine values”. (lines 203-5, Page 9)

“While majority of included studies used Jaffe kinetic method, others lacked information on serum creatinine assessment. Therefore, we could not assess whether the variation in laboratory assessment of serum creatinine contributed to heterogeneity in pooled estimates.” (lines 269-72, Page 12).

---

## [Editor Report · Decision Letter 1]

19 Aug 2021

PONE-D-21-16355R1

Burden of chronic kidney disease in the general population and high-risk groups in South Asia: a systematic review and meta-analysis

PLOS ONE

Dear Dr. Shrestha,

Thank you for submitting your manuscript to PLOS ONE. After careful consideration, we feel that it has merit but does not fully meet PLOS ONE’s publication criteria as it currently stands. Therefore, we invite you to submit a revised version of the manuscript that addresses the points raised during the review process.

You have resubmitted the revised manuscript, and during the preliminary revision I have found that the submitted files are duplicated - there are two "clean" manuscript versions, there are two files for each figure, and two Supplementary files. To facilitate the revision and avoid any confusion, please resubmit the files with only one file for each Item Family (i.e only one "clean" manuscript version, one file for each figure, and one Supplementary file).

It's a technical but important issue. Having these files, it will be possible to proceed with the reviewers involvement.

We look forward to receiving your revised manuscript.

Kind regards,

Boris Bikbov

Academic Editor

PLOS ONE
---

## [Decision Letter · Decision Letter 2]

29 Sep 2021

Burden of chronic kidney disease in the general population and high-risk groups in South Asia: a systematic review and meta-analysis

PONE-D-21-16355R2

Dear Dr. Shrestha,

We’re pleased to inform you that your manuscript has been judged scientifically suitable for publication and will be formally accepted for publication once it meets all outstanding technical requirements.

Kind regards,

Boris Bikbov

Academic Editor

PLOS ONE

Additional Editor Comments (optional):

Reviewers' comments:

Reviewer's Responses to Questions

**Comments to the Author**

1. If the authors have adequately addressed your comments raised in a previous round of review and you feel that this manuscript is now acceptable for publication, you may indicate that here to bypass the “Comments to the Author” section, enter your conflict of interest statement in the “Confidential to Editor” section, and submit your "Accept" recommendation.

Reviewer #1: All comments have been addressed

Reviewer #3: All comments have been addressed

2. Is the manuscript technically sound, and do the data support the conclusions?

Reviewer #1: Yes

Reviewer #3: Yes

3. Has the statistical analysis been performed appropriately and rigorously? 

Reviewer #1: Yes

Reviewer #3: Yes

4. Have the authors made all data underlying the findings in their manuscript fully available?

Reviewer #1: Yes

Reviewer #3: Yes

5. Is the manuscript presented in an intelligible fashion and written in standard English?

Reviewer #1: Yes

Reviewer #3: Yes

6. Review Comments to the Author

Reviewer #1: (No Response)

Reviewer #3: (No Response)

7. PLOS authors have the option to publish the peer review history of their article (what does this mean?). If published, this will include your full peer review and any attached files.

Reviewer #1: No

Reviewer #3: No

---

## [Editor Report · Acceptance letter]

1 Oct 2021

PONE-D-21-16355R2 

Burden of chronic kidney disease in the general population and high-risk groups in South Asia: a systematic review and meta-analysis 

Dear Dr. Shrestha:

I'm pleased to inform you that your manuscript has been deemed suitable for publication in PLOS ONE. Congratulations! Your manuscript is now with our production department. 

Kind regards, 

on behalf of

Dr. Boris Bikbov 

Academic Editor

PLOS ONE